# Automating the Management of 300 Years of Ocean Mapping Effort in Order to Improve the Production of Nautical Cartography and Bathymetric Products: Shom's Téthys Workflow †

**Julian Le Deunf [1,2,*]** , **Thierry Schmitt [2]** , **Yann Keramoal [2]**, **Ronan Jarno [2] and Morvan Fally [2]**

1 IMT Atlantique, Lab-STICC, UMR CNRS 6285, F-29238 Brest, France
2 Service Hydrographique et Océanographique de la Marine (Shom), 13 rue du Châtellier, 29200 Brest, France
* Correspondence: julian.le.deunf@shom.fr
† This paper is extended from conference paper "Téthys: automating a data workflow compiling over 300 years of bathymetric information", San Diego, CA, USA, 20–23 September 2021.

**Abstract:** With more than 300 years of existence, Shom is the oldest active hydrographic service in the world. Compiling and deconflicting this much history automatically is a real challenge. This article will present the types of data Shom has to manipulate and the different steps of the workflow that allows Shom to compile over 300 years of bathymetric knowledge. The Téthys project for Shom will be presented in detail. The implementation of this type of process is a scientific, algorithmic, and infrastructure challenge.

**Keywords:** bathymetric data; quality analysis; data fusion and management; nautical cartography

## 1. Introduction

Since 1720, Shom, the French Hydrographic Service, has collected information describing the physical marine environment, including bathymetric (depth) measurements. These 300 years of data holdings originate from different types of sensors: either through mechanical means (lead-line from early 1800 to the 1920s) [1], or more commonly from acoustic sounding (single-beam since the 1820s, then using multi-beam sounders since the 1980s) [2] or even from optical sounding (lidar since 2005) [3]. The data acquired therefore has different characteristics and qualities. These are at the basis of nautical products, including nautical charts, ensuring the safety of navigation for the mariners, and compliance with Regulation 9 Chapter V of the SOLAS convention; see [4].

All these accumulated data are integrated into a single dedicated database, Shom's bathymetric database (SBDB), see Figure 1, managed as a pile of overlapping and/or intersecting surveys. As of 2022, the SBDB holds over 11,400 surveys. Currently, each cartographic operator that generates nautical charts or digital terrain models must go through a laborious process of selection of bathymetric information.

The Téthys project is an in-house project aimed at constituting Shom's bathymetric reference surface for which source data have been selected in order to generate the most accurate and up-to-date surface, satisfying the criteria related to the safety of navigation.

In this paper, we will first provide elements related to the purpose of bathymetric data fusion along with the current methodology for the production of nautical information. With this context being laid down, we will then explain the details of the Téthys project: notably the data model (especially the geographical extent of the data) and the semi-automated processing chain. Finally, through examples, we will present the current production status results, before proposing conclusive perspectives.

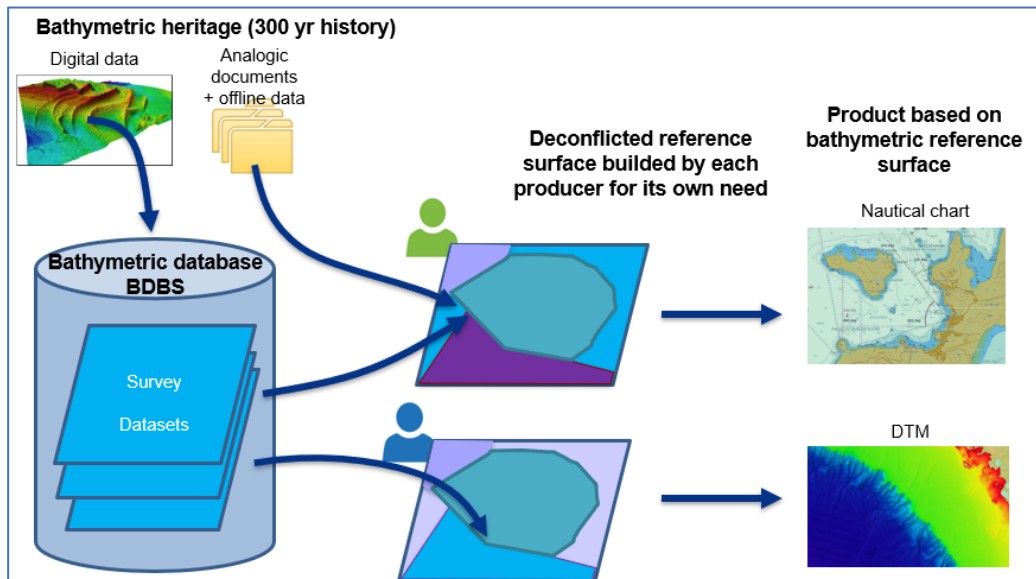

**Figure 1.** General operation before the Téthys project, where each operator recovers all the data before processing it according to his needs.

## 2. Bathymetric Data Fusion State of the Art

Bathymetric data fusion is a vast research topic, which has been largely discussed in the literature [5–8]. Globally speaking, the process deals with the aggregation of depth soundings for dedicated products. Depth soundings can originate from diverse sensors (acoustical: single beam or multibeam; estimated from active or passive remote sensing technologies: Lidar, satellite-derived bathymetry, Radar Altimetry) and prior information originating from historical digitized charted products or ENCs. These sources of data often combine multiple levels of vertical and horizontal accuracies; they integrate various levels of data density (from sparse coverage to tenths to a hundred of soundings per m$^2$) from different ages and different processing levels (from raw soundings to fully corrected and precisely vertically reference datasets associated with a characterization of the associated level of confidence).

Due to the scarcity and the difficulty of collecting bathymetric data, the users of bathymetric products generated through a process of data fusion are often aware that data with mixed levels of quality is the best that can be achieved. The scale and coverage of the generated product are often the key factors associated with the level of refinement and effort to be put into the compilation process [9]. At a small scale (i.e., roughly speaking above 1/10.000.000), common current practice is to aggregate all the sources and use deterministic interpolators (quite often spline-in-tension). This is exemplified by a worldwide digital terrain model such as SRTM15+ [10] or GEBCO [11]. Strong limitations arise, notably when the level of details and accuracy sought is higher (larger scale, higher resolution) in particular in dynamic areas (e.g., sandwaves).

For this kind of context, a selection process, focusing principally on the most recent data sources, is needed to prevent an unrealistic representation of the seabed. Regional efforts such as EMODnet Bathymetry incorporate this step in their methodology [12]. In shallow water areas, where vertical precision is essential, attention must be paid to the quality of the dataset with respect to vertical accuracy, with particular attention being paid to the tidal referencing methodology [13]. Some national data compilation efforts well illustrate this further effort in the data selection and management process [9,13,14].

Moreover, when it comes to bathymetric product usages where the security of navigation is at stake, all the previous steps of the selection processes are taken with extra caution, as much as the representation of the various confidence levels associated with the source data is included in the process of compilation. In this sense, the CATZOC (Category of the Zone of Confidence) is an accompanying layer of the navigational chart [15] describing

the data uncertainty [16], which has as much importance as the depth information layer, and which is supported by dedicated and well-managed metadata associated to the source datasets (see Section 4).

A major keystone in the reasoning for generating bathymetric products from hydrographic data is the concept of "Navigation Surface," also known as reference surface, first introduced by Smith [17]. The navigation surface is a bathymetric surface product made up of a collection of sources assuring that items critical for navigation are preserved. One of the benefits of this concept is to locally ensure the selection of the most relevant data sources compatible with the most stringent use (here considered to be the safety of navigation).

The process of ensuring the suitable selection of the local knowledge is also known as "deconfliction," which is a decision-making process whereby a selection of the sources of data is made from the best representation of the physical coverage (2D polygon area) and associated information such as, for example, the age and reliability of the source. This process is either undertaken (see Figure 2) through:

- a decision of "remove/restore," where one of two (or many) overlapping sources uptakes the others.
- a decision of "supplement," where the datasets under concern are merged without giving priority (or weight) to one dataset over the other.

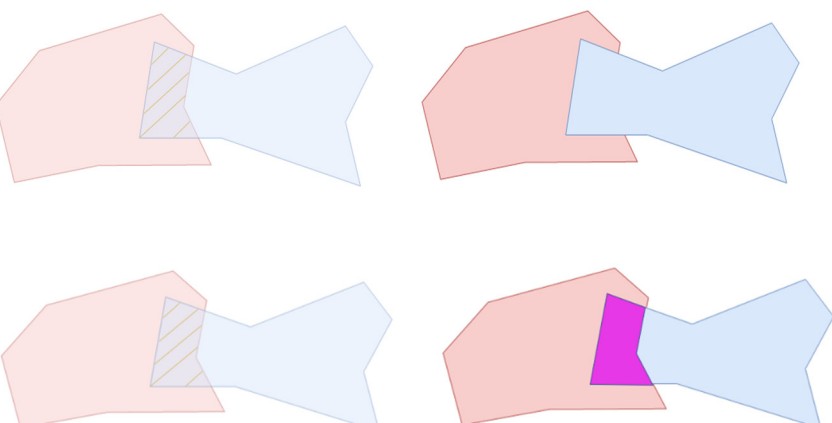

**Figure 2.** Deconfliction process representation. **Top**: a remove/restore decision where the blue survey uptakes the red one; **Bottom**: a supplement decision where both surveys are merged.

By creating such a navigation surface at the geographical scale of the survey (highest scale), the objective is to undertake the decision-making selection process once for all sthe cales. In doing so, gains are already measured in generating bathymetric products (nautical charts and digital terrain models) by capitalizing on the selection efforts while strengthening the management and valorization of the source information. Currently, other similar national initiatives are underway, such as BlueTopo from the NOAA [18].

## 3. Qualitative Description of Data and Metadata

The information used in the production of Shom's reference surface is composed of bathymetric data from hydrographic surveys (sounding point clouds in the form of x,y,z triplets or bathymetric raster surfaces) to which are associated spatial metadata representing the extent of the survey (minimal enclosing surface, later defined as MES and SME in French) and a series of attribute metadata.

The attribute metadata are associated with internationally recognized metadata from the IHO S-57 standard [19,20], such as CATZOC (Category Zone of Confidence), POSACC (Position accuracy), SOUACC (Sounding accuracy); see Figure 3 as well as internally defined attributes. From the latter, we can cite examples such as the Codval attribute, which indicates the validity/invalidity of a bathymetric survey, or the Captur attribute, which indicates the type of bathymetric sensors used at sea for its acquisition. Within the

framework of the Téthys, the attribute metadata are managed in a conventional eXtended Markup Language (XML) format with a key-value formalism.

```xml
<?xml version="1.0" encoding="UTF-8"?>
<BDB_Simple_Attributes>
  <Attribute name="CATZOC">
    <Value>zone of confidence B</Value>
  </Attribute>
  <Attribute name="NINFOM">
    <Value>20210611 : STM/BATHY/GDBS Reprise de la SME. Création d'un fichier &#13;
Jun 11, 2021 8:22:55 AM&#13;
Bathymetry replaced by blecorre</Value>
  </Attribute>
  <Attribute name="OBJNAM">
    <Value>S201500100-1</Value>
  </Attribute>
  <Attribute name="POSACC">
    <Value>2.1</Value>
  </Attribute>
  <Attribute name="RECDAT">
    <Value>20170216</Value>
  </Attribute>
  <Attribute name="SOUACC">
    <Value>0.43</Value>
  </Attribute>
```

**Figure 3.** Classic metadata file with S-57 [19] attributes.

The spatial metadata of a bathymetric survey is meant to represent the area of the seabed that has been recognized following the acquisition/processing stage. Fidelity to the actual coverage of the dataset is key as this polygon will be used as part of the deconfliction process. In order to be the most representative, three conditions on the relationship between the minimal enclosing surface polygon and its associated soundings have been defined and must be verified:

- Unicity: Each sounding of the dataset is included in a single MES.
- Density: Soundings that have a distance with the nearest neighbors less than 5 times (defined from the hydrographic expertise, Case 1 of Figure 4) the intrinsic resolution of the acquisition sensor is gathered in the same MES. Otherwise, a new polygon is created (Case 2 of Figure 4). Eventually, the sounding is considered as an isolated sounding if it is impossible to aggregate it with its neighbors (Case 3 of Figure 4).
- Representativeness: The contour (internal and external) of the generated multi-polygons is buffered with a distance depending on the characteristics of the survey: horizontal uncertainty and intrinsic resolution. This is a sensitive point to avoid removing a shoal at the border of the survey with a too-loose MES when the area has not been strictly covered.

Figure 4 schematizes all these criteria. Note that following this process, a single survey is represented by a single or a multi-polygon also including holes in their geometry.

Prior to 2018, the MES was constructed manually by operators at Shom, as no known algorithm provided satisfaction (representativity and computing performance). Moreover, this tedious work was also subject to operators' biases, which strongly motivated the development of an automated MES envelope generation.

In order to determine the most accurate spatial coverage for a bathymetric survey, we first studied the $\alpha$-shape algorithm, which is a classical computational geometry method [21] that is a refinement of the convex hull method [22]; both are available in numerous GIS solutions (e.g., QGIS) based on a Delaunay Triangulation followed by an analysis of the length of the triangle edges and suppression of the triangle edges, the lengths of which are above the defined $\alpha$ length. The algorithm has a complexity of $O(n \log n)$, with $n$ representing the number of points. The $\alpha$-shape algorithm is used to obtain the line segments composing the perimeter of a set of points in the plane, thus allowing the building of the strictest spatial boundary from these segments composing the boundary of the input point cloud. The key parameter of this algorithm is the $\alpha$ value, which defines whether a

segment will be considered a right-of-way boundary or a core segment. Nevertheless, this method has strong limitations, with changing density of the point cloud, considering the static definition of the α parameter.

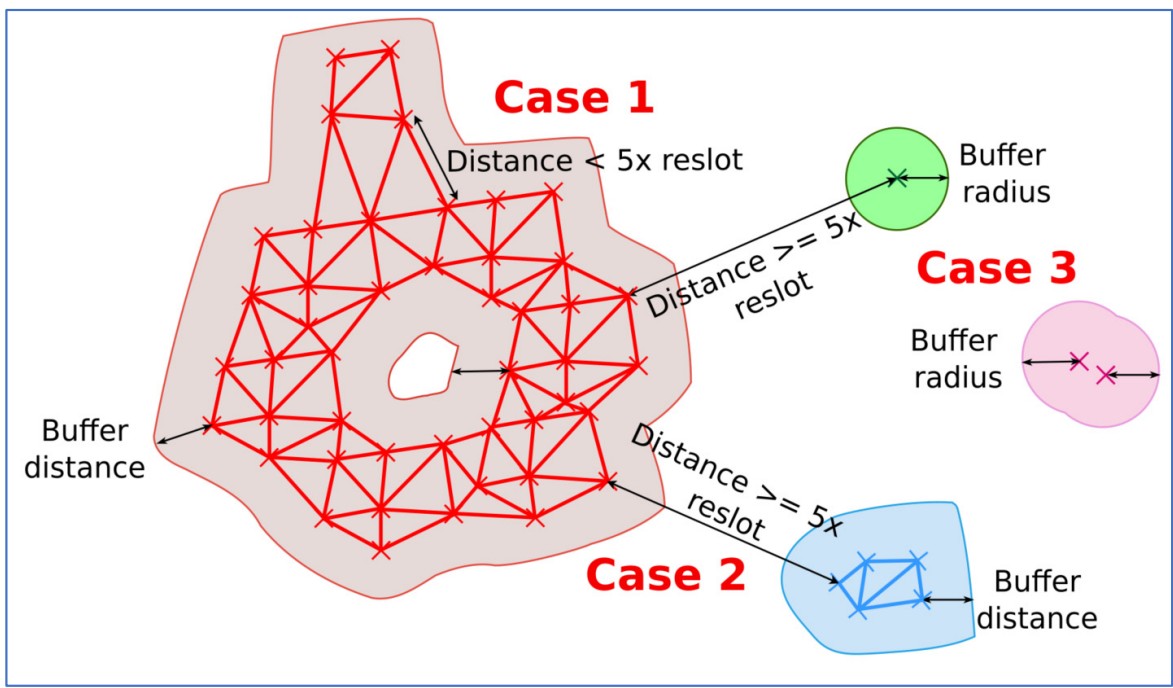

**Figure 4.** Shom MES definition.

In order to alleviate the limitation introduced by the α-shape algorithm and to optimize the searching phase of the points located at the border of the survey, we introduced the QuadSME algorithm [23]. Based on the bathymetric point cloud and its horizontal uncertainty (the POSACC), the algorithm had five steps:

- The first step of this methodology was the import of the data in the form of a point cloud including triples (x, y, z) and the value of the associated POSACC.
- The second step of the methodology consisted of a geospatial indexation based on a first quadtree segmentation [24], with the number of points per quadrant set to 5 million points as the stopping criterion.
- The third step of the methodology consisted of a second quadtree indexing. The space was divided to keep only quadrants validating either a density criterion or a maximum number of soundings (arbitrarily defined at 1000). The density criterion corresponded to the number of points in each sub-quadrant constituting a main quadrant. If the density was identical (judged by a threshold) for each child quadrant, then the parent quadrant was considered as homogeneous.
- The fourth step of the methodology consisted of the generation of polygons containing the soundings. First, a characteristic resolution of the point cloud included in the sub-quadrant was calculated to adapt to the potential differences in density of the input point cloud. Then, partitioning and detection of isolated points were performed. The objective was to build specific envelopes for the isolated points and build clusters of points with the same density before creating the polygons. Finally, a Delaunay triangulation was performed on the different clusters and the associated polygon was extracted.
- The fifth and last step of the methodology consisted of dissolving the polygons generated during the previous steps to form the final MES. The geometries were merged via a process of dilation/erosion (creation of a buffer) of the geometries to

remove construction holes, see Figure 5 which represents a generated MES and the associated quadtree decomposition.

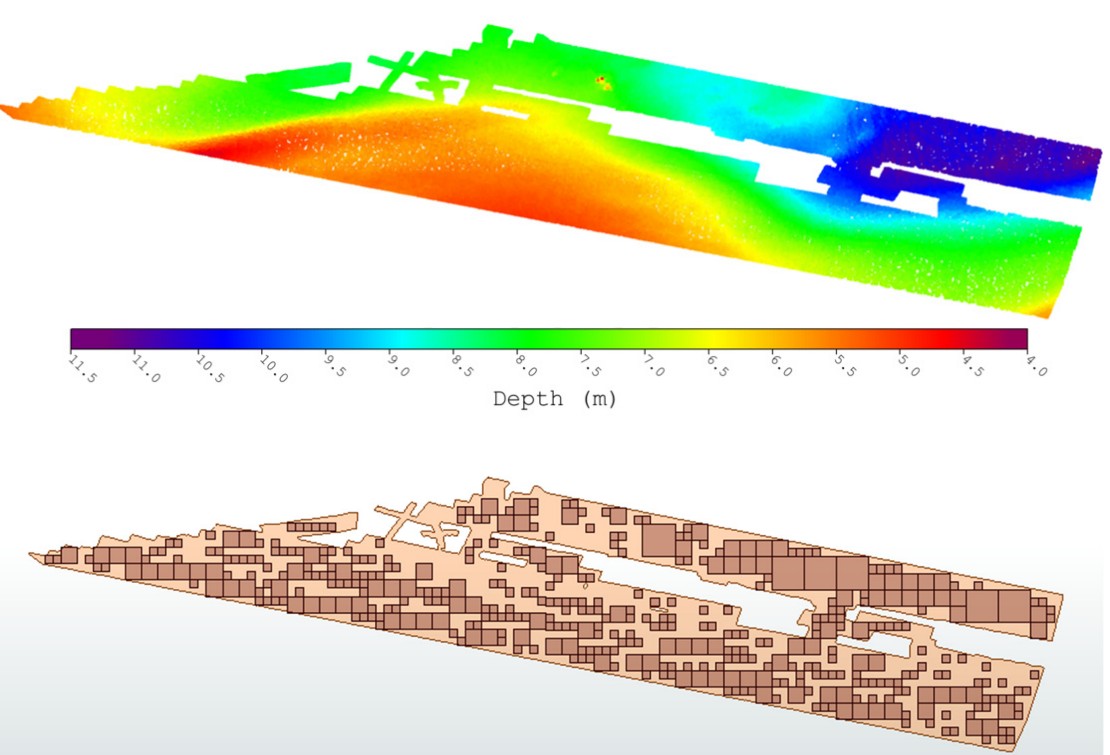

**Figure 5. Top**: a bathymetric survey (point cloud—color according to depth value); **Bottom**: the MES generated by the QuadSME algorithm with the preserved holes. Note also the corresponding quadtree decomposition.

In order to compare the representativity of the geometries both generated by the $\alpha$-shape and QuadSME methods, the Haussdorf-Pompeiu [25] distance metric was selected. The Hausdorff-Pompeiu distance is a topological tool that measures the distance between two subsets of a metric space. It was therefore very suitable for comparing the maximum distance between two spatial areas, which allowed the dissimilarity of the two shapes to be measured. From Table 1, which shows the results associated with five surveys differing in size and geographical coverage, it can be observed that the Hausdorff-Pompeiu distance for the QuadSME method was always smaller and therefore more faithful to the reference $\alpha$-shape method. Also, a fact to be noticed is that the distance value remained in the same order of magnitude for lots with few soundings (first two examples of Table 1). Moreover, for larger size datasets (last three examples of Table 1), the QuadSME method provided a Hausdorff-Pompeiu distance better within one order of magnitude.

**Table 1.** Computation of the Hausdorff-Pompeiu distance (in meters) for five bathymetric surveys.

| Survey Name | Soundings Number | $\alpha$-Shape Distance | QuadSME Distance |
|---|---|---|---|
| S202099900-001 | 2743 | 165.6 | 141.9 |
| S201207000-5 | 25,718 | 86.7 | 76.5 |
| E201804100-002 | 128,939 | 16.6 | 0.9 |
| S202102500-001 | 1,070,131 | 182.9 | 25.1 |
| S200701200-1 | 10,829,541 | 1340.3 | 118.3 |

The computation time associated with each method was also compared, using the same computing facility (Intel Xeon 6248 2.50 GHz, 32 Gb RAM). The QuadSME method,

see Figure 6, showed better computation times than the QuadSME algorithm compared to the α-shape algorithm, especially when the number of points was greater than one million. For a number of soundings of the order of magnitude of 10 million, the QuadSME method was 40 times faster than the α-shape algorithm, most likely because of the quadtree partitioning (Steps 2 and 3). Processing time was further improved by multiprocessing the QuadSME method, with operations from Steps 2 to 4 performed independently and in parallel on each of the quadrants.

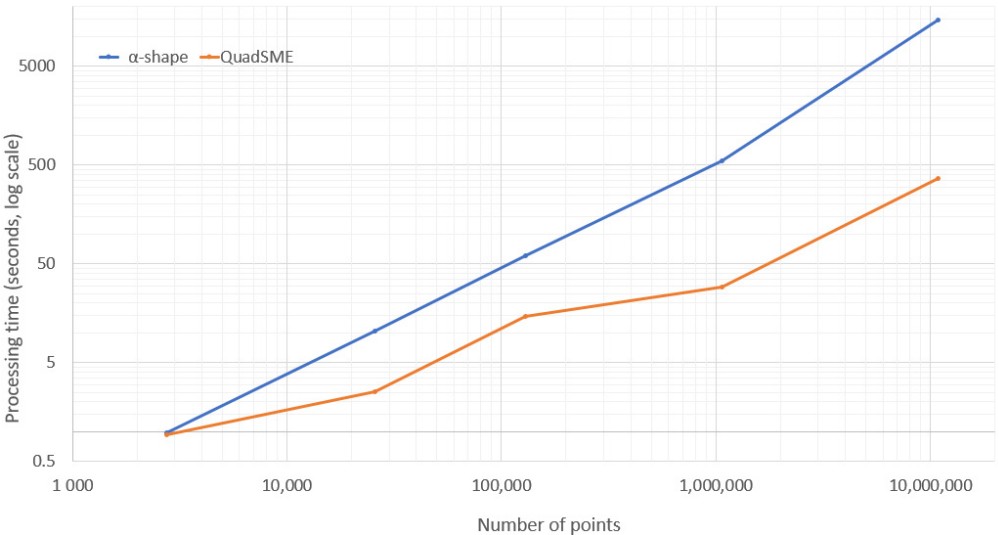

**Figure 6.** Computation time for α-shape and QuadSME algorithms.

On the other hand, the computation time of the QuadSME method was very dependent on the homogeneity of the distribution soundings. Thus, when the density criterion was quickly reached then the quadtree process stopped. Conversely, when the sounding distribution was not homogeneous in the sub-quadrants or when the data contained many holes, then the computation time was longer because it was necessary to go to the end of the quadtree decomposition.

## 4. Téthys Workflow

Following a detailed and accurate representation of the source dataset, as described in the previous section, the deconfliction process was wisely undertaken, leading to the generation of the bathymetric surface reference. The overall workflow, see Figure 7, was carried out according to the following processes:

- From the different original surveys, verification of all data and metadata content was performed, benefiting the SBDB consistency directly.
- The conflicts between the superimposed datasets were resolved according to the qualitative elements carried by the metadata (hydrographic qualification, ages, etc.).
- The compilation (combination or cutting/replacement) of the data was undertaken following the priorities previously defined between the datasets in Step 2.

Considering the vastness of the French exclusive economic zone (EEZ), this workflow was operated on 1° by 1° geographic tiles. More than 300 expert rules validated by Shom hydrographers and cartographers were implemented in this process.

The Téthys project offers each operator data where their interactions are validated and verified by a set of attribute rules and priority constraints related to each other. The resulting surface is directly exploitable, without any particular expertise, and is reproducible.

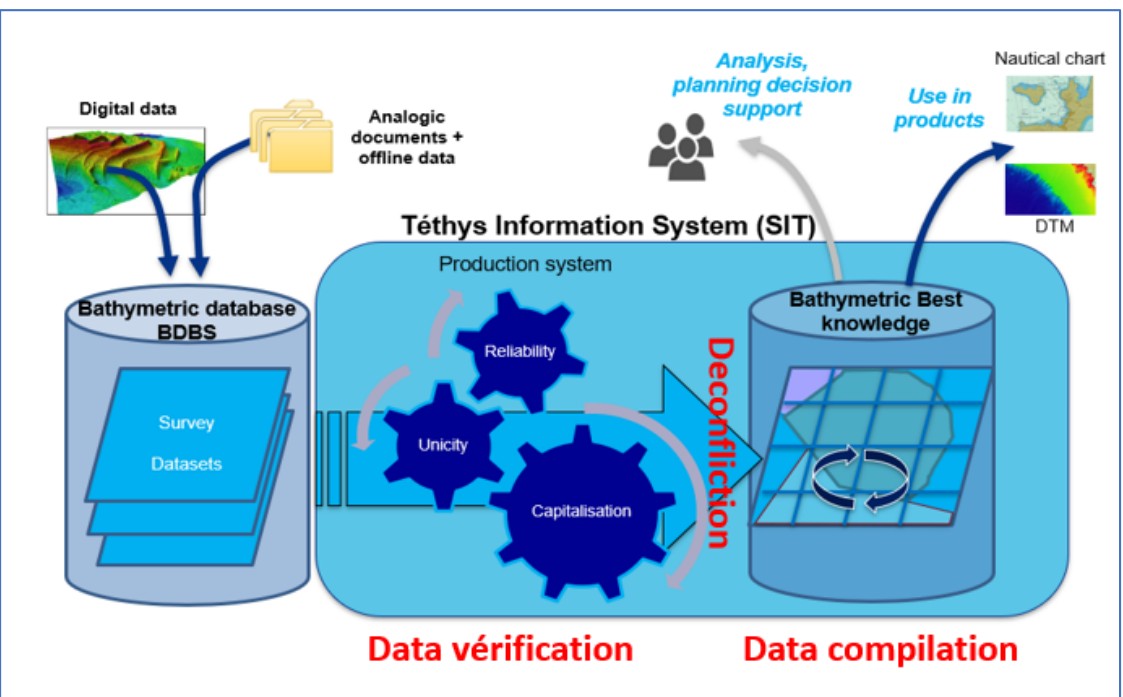

**Figure 7.** General workflow of Téthys Project, where the Téthys base is the reference bathymetric bottom.

Automation of this workflow can be implemented based on several technologies that best handle open and proprietary geospatial data formats, along with efficient manipulation of large volumes of data. Fundamental to this implementation are the use of:

- Extract Transform Load (ETL) software handling spatial information: The dedicated FME software [26] supports geospatial data extraction (Extract) from homogeneous or heterogeneous sources, followed by the processing stage (Transform) of the data into a proper storage format/structure; and, finally, the data is loaded into a dedicated target database. In addition to data transformation tools, spatial ETL solutions also contain various geoprocessing algorithms to process and analyze spatial and non-spatial data (e.g., geometry validation and repair, topology check, or creating and merging attributes, etc.). The software allows this tool to have several advantages for the needs of Téthys. Figure 8 illustrates the no-code graphical FME interface, based on multiple data-driven interactors. Such a workflow processing environment facilitates development and subsequent maintenance.
- Direct geo-processing in a dedicated working database via SQL scripts: The choice was made to use the combination of a PostgreSQL/POSTGIS database, overlaid with the pgPointCloud extension [27]. This environment benefits from the adapted geospatial point cloud indexing capabilities commonly used for the management of large LIDAR point clouds, which have similar characteristics to bathymetric soundings. Note that direct interaction with the pgPointCloud data structure is managed through the PDAL library [28].
- Dedicated APIs to allow for the manipulation of proprietary format. The current SBDB is currently managed under the proprietary software Teledyne CARIS Bathymetric DataBase, and Python bindings built upon a dedicated API provided by the software manufacturer [29] allow for the transformation into open and interoperable formats.

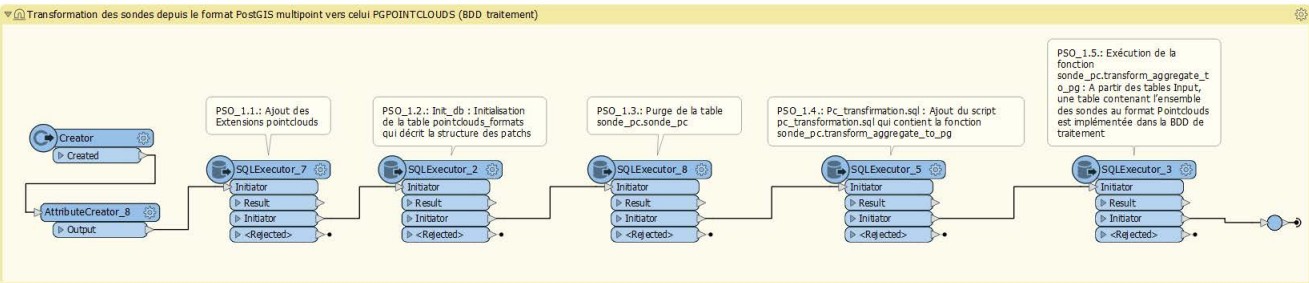

**Figure 8.** Example of FME interface performing the processing to generate the surface reference.

The first deconfliction performed is shown in Figure 9 which distinguished the stages before and after this process; each color represented the MSE of a survey. On this first tile (called 145_81, a name inspired by the Marsden square [30]), 115 surveys were used as input data and 441,418,088 associated soundings were processed. At the end of the processing chain, only 96 surveys were finally retained and 310,970,981 soundings were integrated into the Téthys. Of these bathymetric data, over 6000 soundings were digitized from old nautical charts.

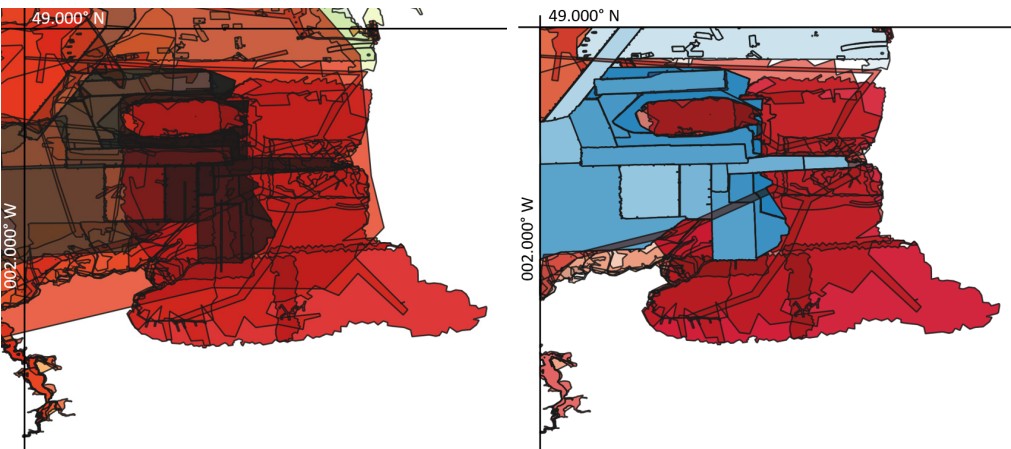

**Figure 9.** First tile of the Téthys project: **on the left**, surveys studied for deconfliction; **on the right**, surveys cut and kept after the deconfliction process (we distinguish easily the remove/restore or supplement decision especially in the blue part on the right side).

The area covered extends from the port of Saint-Malo in the west, to the bay of Mont Saint-Michel in the east, and from the south of the Rance to the town of Coutance in the north. The result of this deconfliction process raised 44,167 conflicts between intersecting data sources. Quality control of the tiles was performed by comparing, among other things, previously generated navigation products, such as the official Electronic Navigational Chart (ENC). This first work has recently benefited cartographers who published the nautical chart covering the Chausey Islands and the production of the topo-bathymetric DTM, which covers the approaches to Saint-Malo; see [30].

Shom agents have access to these bathymetric data via an internal geographic web portal. Selected layers can be queried, filtered, and downloaded in well-known GIS formats (ASCII, shapefile, GeoPackage). Bathymetric data are extracted by defining a bounding box of the area of interest. Users can also load web services (WMS, WFS, WCS) or GIS vector data into the portal.

## 5. Discussion and Perspectives

The current concept underlying the use of the Téthys is oriented towards cartographic use applied for the safety of navigation, which translates into the implementation of more than 300 expert rules to ensure the control and deconfliction of bathymetric surveys.

However, different concepts of use might require different rules or preferences to be implemented in the deconfliction process. For example some users, with fewer constraints on the selection of shoals, but stronger constraints on the statical robustness of the bathymetric information (digital terrain elevation surfaces for the use of oceanographic modeling) might welcome relaxed deconfliction rules with the potential weighting of the prioritized sources of overlapping surveys [7,31]. It would be relevant to look at the expert needs concerning the deconfliction rules to be implemented in order to adapt the current workflow to these new practices.

Moreover, with an increasing effort being brought to the automation of the overall workflow, the transformation of the hydrographic profession is questionable. While an effort to generate the first iteration of the reference tiles is currently needed, it is also believed that, through subsequent updates, the hydrographers will have to focus on more and more specific technical issues related to their training without being distracted from minor processing tasks; hence generating a virtuous cycle.

The Téthys workflow systematically implements automation techniques and methodological developments that allow it to take advantage of the intelligence of the data. The generation of the surface reference based on the most relevant bathymetric knowledge allows selected information to be effectively and efficiently provided as support for the generation of marine charts. This methodology and its implementation can prepare the French National Hydrographic to meet the challenges of the future as it better manages bathymetric data, makes it more efficiently usable for end-products, and considers the diversity and increasing volume of bathymetric data to be handled in the close future.

The target is to model all the tiles in the French metropolitan EEZ by the end of the first quarter of 2024; at the time of writing this paper, more than 46% of this area is already produced. Furthermore, updating the bathymetric reference navigation surface with the most up-to-date surface and new incoming surveys, is a crucial task that is easily enabled by the Téthys process.

**Author Contributions:** Conceptualization, R.J., M.F., J.L.D., T.S. and Y.K.; methodology, R.J., M.F., J.L.D., T.S. and Y.K.; software, R.J., M.F., J.L.D. and T.S.; writing—original draft preparation, J.L.D.; writing—review and editing, J.L.D. and T.S.; supervision, Y.K.; project administration, Y.K.; funding acquisition, Y.K. All authors have read and agreed to the published version of the manuscript.

**Funding:** This project, financed by the Fonds de Transformation de l'Action Publique, was a structuring project for the Shom. It responded to the need to structure bathymetric information by generating a surface from information from sensors and different acquisition methodologies, and capitalizing on decisions made on it.

**Institutional Review Board Statement:** Not applicable.

**Informed Consent Statement:** Not applicable.

**Data Availability Statement:** Data are not available due to commercial and government restrictions.

**Acknowledgments:** We would like to extend our gratitude to the Bathymetry Department for all their efforts to complete this project and to the company Geofit for the web portal development.

**Conflicts of Interest:** The authors declare no conflict of interest.

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
