# Peer review of "Automating the Management of 300 Years of Ocean Mapping Effort in Order to Improve the Production of Nautical Cartography and Bathymetric Products: Shom’s Téthys Workflow†"

_2673-7418, doi:10.3390/geomatics3010013_

Round 1

Reviewer 1 Report

The work is devoted to the development of marine cartography and bathymetric datasets production automation according to various data sources, accumulated over a three-hundred-year period. The presented results are relevant and significant, the work is written in a clear language, much attention is paid to details, and the presented diagrams well explain the described processes.

Several comments can be made on the content of the manuscript:

1. For a better understanding of figures 8 and 10, it is recommended to apply a coordinate grid on them. 

2. In Figure 8, the lower parts of the letters are cut off.

3. It is not clear from the caption to Figure 8, which surveys highlighted in different colors are meant. It is recommended to add an appropriate clarification.

4. The presentation of Figure 10 seems unclear - what exactly does color intensity mean in each part of the figure? Is it possible to add a colorbar? 

5. It is recommended to decipher the abbreviation EEZ (line 249)

6. 32 Gb RAM (Line 221)

Author Response

Dear reviewer,

Thank you very much for your comments and suggestions.

We have carried out the proposed corrections in particular (see word attached):

  • Points 5 and 6 have been dealt with directly in the text;
  • Figure 8 has been removed as it did not provide any additional information to the article;
  • We add a coordinate grid to figure 9 (previously figure 10);
  • We add an explanation in the body of the text and a better colour for figure 9 (previously figure 10).

Thanks again and have a nice day,

Julian Le Deunf.

Reviewer 2 Report

This work is nicely links a long mapping tradition with the future needs of the maritime domain and the hydrographer's profession with the technical progress.

Some improvements on the following figures are suggested:

Figure 4 - to label all 3 cases for the Density criterion (e.g.  case 1: distance less than 5 times; case 2: distance more than 5 times; case 3: isolated soundings). The preceding text should be revised accordingly. Referring to Figure 4 and numbering the cases as 3.1 and 3.2 might be confusing.

Figure 5 - on the bottom, to represent thematically the depth values of the MSE (in accordance to the color grading of the bathymetric survey ones)

Figure 7 - to label the 3 steps on the figure (e.g. 1. data verification; 2. conflicts resolution (or deconfliction); 3. data compilation)

Author Response

Dear reviewer,

Thank you very much for your comments and suggestions.

We have carried out the proposed corrections in particular (see doc attached):

  • Label all 3 cases for the Density criterion for figure 4 ;
  • Add a color bar for the figure 5 ;
  • Add 3 step label for the figure 7.

Thanks again and have a nice day,Julian Le Deunf.
